Extended Abstract Track

# Representing Repeated Structure in Reinforcement Learning Using Symmetric Motifs

**Matthew J. Sargent**                               matthew.sargent.19@ucl.ac.uk
**Augustine Mavor-Parker**                    augustine.mavor-parker.15@ucl.ac.uk
**Peter J. Bentley**                                        peter.bentley@ucl.ac.uk
*Centre for Artificial Intelligence, University College London*

**Caswell Barry**                                          caswell.barry@ucl.ac.uk
*Department of Cell and Developmental Biology, University College London*

**Editors:** Sophia Sanborn, Christian Shewmake, Simone Azeglio, Arianna Di Bernardo, Nina Miolane

## Abstract

Transition structures in reinforcement learning can contain repeated motifs and redundancies. In this preliminary work, we suggest using the geometric decomposition of the adjacency matrix to form a mapping into an abstract state space. Using the successor representation (SR) framework, we decouple symmetries in the transition structure from the reward structure, and form a natural structural hierarchy by using separate SRs for the global and local structures of a given task. We demonstrate our method can achieve high policy evaluation accuracy while using representations that are significantly compressed.

**Keywords:** symmetry, reinforcement learning, successor representation

## 1. Introduction

Knowledge of a task's structure can enable more efficient policy learning in reinforcement learning (RL) Ravindran and Barto (2001). For example, policies can be constructed that are temporally hierarchical Sutton et al. (1999) or an environment's state space can be reduced by recognising its symmetries van der Pol et al. (2020b). Unfortunately, discovering an environment's latent structure is difficult. Existing methods usually find reduced state-action spaces by learning a global mapping between the environment's Markov decision process (MDP) and a more abstract MDP van der Pol et al. (2020a); Mavor-Parker et al. (2022). In this paper we use a different strategy—we use results from graph theory and network analysis MacArthur et al. (2008); Sánchez-García (2020) to isolate repeated structural motifs in an MDP, allowing for the compression of its transition function into a more compact form. Additionally, our approach is formulated using the Successor Representation (SR) Dayan (1993), meaning structural regularity can be decoupled from the reward function. Empirically, we demonstrate effective compression of SRs formed over small world graphs Barabási and Albert (1999); Watts and Strogatz (1998), opening up future avenues of work using hierarchical reinforcement learning approaches within the SR.

## 2. Related Work

To operate in large state spaces, knowledge of the structural regularities of an environment is essential. Traditionally structural knowledge is obtained implicitly by learning an intermediate representation useful for value judgements—or explicitly by learning a compressed latent model of the environment Hafner et al. (2019). Other approaches encode inductive biases into policies about structural regularities Ravindran and Barto (2001). For example, van der Pol et al. (2020b) build knowledge of global symmetries into policy networks, while van der Pol et al. (2020a) learn the equivariances inherent in the dynamics of a given MDP. The SR is a biologically plausible middle ground between model-based and model-free reinforcement learning that contains information about an environment's structure—decoupled into its reward function and expected transition dynamics Dayan (1993); Stachenfeld et al. (2017). However, as far as we are aware, we are first to build an abstraction on top of the SR to compress it into a more compact form, which we describe next.

## 3. Geometric Decomposition

For a given MDP, the transition function $T(s, a, s')$ implies an adjacency graph $G = (\mathcal{S}, E)$ where $\mathcal{S}$ is the set of states in the MDP, and $E$ is the set of edges: pairs of states $(s, s')$ that are connected under some action. Only deterministic MDPs are considered. Structural regularities in this graph can be described using the notion of *graph automorphisms*. An automorphism is a permutation of nodes that preserves the adjacency structure of said nodes. The set of nodes being permuted is known as the orbit. Within the overall adjacency graph of the MDP, there may exist smaller subgraphs that are symmetric and therefore can be described by some automorphism that permutes their nodes within the bounds of the local symmetry. These automorphisms are often repeated across the structure of a graph—they are hence referred to as motifs (MacArthur et al., 2008). From a given automorphism group, a graph can be represented as the union of the vertices not belonging to any symmetric subgraph, $\mathcal{S}_0$, and the the vertices of each motif *i.e.* $\mathcal{S} = S_0 \cup S_{M_1} \cup S_{M_2} \ldots \cup S_{M_m}$ where $S_{M_i}$ is the subgraph of vertices that are symmetric under a specific automorphism. This decomposition is the geometric decomposition of a graph (MacArthur et al., 2008). The automorphism group for a given graph is not unique; we follow (Sánchez-García, 2020) and generate subgraphs that are non-overlapping, and therefore automorphisms that are support-disjoint. We use `saucy` (Darga et al., 2012) to generate our automorphism groups, and in-line with (Sánchez-García, 2020) we restrict the motifs found to two orbits in size.

## 4. Structural Hierarchies with Local Symmetry and Successor Representations

Once an automorphism group is generated and the geometric decomposition of the graph is found, we generate a reduced graph with adjacency matrix $\hat{A}$ by representing every occurrence of a symmetric subgraph with a single node, whilst also keeping an index of which node represents which subgraph and the local transition structure of each symmetric subgraph $A_i$. An illustrative diagram of this can be found in 1. For the purposes of this work, we will only consider the task of policy evaluation under a random walk policy. If one were to sample trajectories from this graph following a policy, or compute a policy

Extended Abstract Track

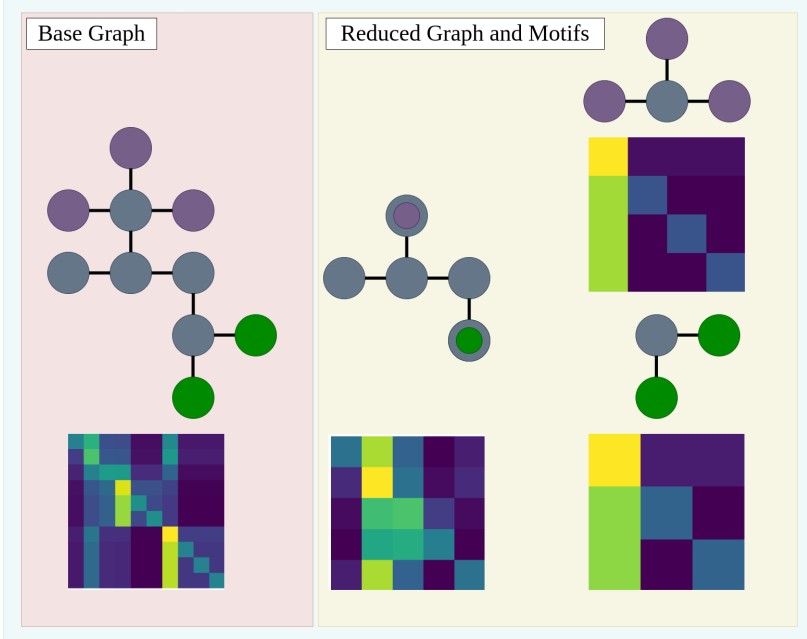

Figure 1: A diagram showing an example graph, and its decomposition into a irreducible core and subgraphs. States that are symmetric under an automorphism are displayed in purple or green; the core is shown as grey. Each graph is shown with a pictorial representation of its random walk successor representation matrix.

through sample-based policy improvement, the graph-subgraph structure would need to be represented hierarchically using the options framework (Sutton et al., 1999), where nodes representing a symmetric subgraph would also lie in the initiating set of an option that would allow the agent to execute a policy specified for the subgraph.

A random walk SR is generated for the reduced graph $\hat{A}$, and for each symmetric subgraph $A_i$. Let $\mathcal{S}_{A_i}$ represent the state space of subgraph $A_i$. We generate expected random walk state transition dynamics using the analytic form of the SR: $\psi^\pi = (\mathbb{I} - \gamma \mathbf{A})^{-1}$, where $\pi$ is the policy, $\psi$ is the SR and $\gamma$ is the discount factor. A policy over the MDP is defined for $m$ subgraphs as $\pi(a|s) = \mathrm{argmax}_a (\psi^\pi(R(s))$— $\psi$ is either the SR of the reduced graph, or a subgraph depending on the state being evaluated. $a$ represents actions and $s$ represents graph states. When evaluating states in $\mathcal{S}_{\hat{A}}$, $R(s)$ for states representing compressed subgraphs is the mean of the rewards contained in the subgraph. There is an inherent bias in estimating the value function in this way, as the state occupancies obtained for $\mathcal{S}_{\hat{A}}$ will not account for occupancies accumulated inside the subgraphs. Future work will study deriving a bound for this value approximation error, but empirically this is not a significant malus.

### 4.1. Results

To test this approach, 100 Barabási-Albert (Barabási and Albert, 1999) and 100 Connected Watts-Strogatz (Watts and Strogatz, 1998) graphs were generated, each with 100 nodes.

# Extended Abstract Track

The Barabási-Albert graphs were generated with $m = 1$ edges preferentially reattached, and the Watts-Strogatz graphs were generated with nodes initially attached to $k = 2$ neighbours and rerouted with $p = 0.33$. The Watts-Strogatz graphs were more densely connected than the Barabási-Albert graphs. We used `networkx` (Hagberg et al., 2008) to generate these graphs. To convert these to MDPs, an action space was chosen arbitrarily with size equal to the degree of the most connected node in a graph. For each graph, the value function under a random walk policy was found for 100 reward functions, where the reward function $r(s, a, s') := r(s') := \mathcal{N}(0, 1)$; that is, 100 reward functions for each of the 200 graphs.

The exact value functions can be recovered within symmetric motifs using their random walk SRs and the rewards associated with each state. In the reduced graph, there will be a discrepancy between the value functions as the state occupancy for states in the core will not reflect transitions made inside motifs. As such the difference between the optimal greedy policies obtained by the reduced random walk SR combined with a given reward vector, and the true optimal policies found using the SR defined over the original state space, was quantified as the total variation $TV := \frac{1}{|\mathcal{S}'|} \sum_{s \in \mathcal{S}'} \frac{1}{2}||\pi(s) - \pi^*(s)||_1$. The TV gives an indication of, on average, the proportion of actions taken under the inferred policy that deviate from the optimal policy. The numbers quoted in Table 1 correspond to a deviation of around 4% and 0.1% for the Barabási-Albert and Watts-Strogatz graphs respectively. The mean squared error between the derived value and optimal value is also shown—in practice this is dominated by a small number of graphs per node that are massively overestimated. The compression is shown as a percentage size of the resulting SRs compared to an SR defined over the original graph. For a graph with $n$ states, the number of elements required for storage is equal to $(n - (\sum_i |S_{A_i}| - 1))^2 + (\sum_i |S_{A_i}| + 1)^2$. An additional 1 is added to the second term to allow for a virtual state to be represented in the SRs of the subgraphs that allows for an agent to leave the subgraph and return to the main graph. This is an upper bound and assumes that every subgraph is generated according to a unique motif.

Table 1: Summary statistics of evaluation of the random walk policy over 100 graphs with 100 rewards functions per graph

| Graph Generator | $\mathbf{TV}(\pi, \pi^*)$ | $\lVert V, V^* \rVert_2^2$ | Compression |
|---|---|---|---|
| Barabási-Albert | 0.0231±0.0074 | 19.4±0.8 | 33.6%± 7.7% |
| Watts-Strogatz | 0.0015±0.0001 | 22.1±1.2 | 94.3%± 3.5% |

## 5. Future Work

An additional improvement that can be made is the use of the internal connectivities to adjust the SR of the $\hat{A}$ to reduced the value approximation error. Sánchez-García (2020) use this information in the form of their *Basic Symmetric Motifs* to adjust pairwise network measures over compressed graphs. The largest constraint is the need to know the full adjacency matrix—this is not tractable in reinforcement learning with high dimensional

observations. Instead, we are interested in using the knowledge of symmetric graphs to allow agents to form expectations of local structures given their prior experience.

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
