# OpenReview forum: "Representing Repeated Structure in Reinforcement Learning Using Symmetric Motifs"
_NeurIPS.cc/2022/Workshop/NeurReps — NeurReps 2022 Poster_

### Official Review · Reviewer_FATx · 2022-10-11

**Confidence:** 3
**Soundness:** 3
**Presentation:** 3
**Contribution:** 3
**Overall Rating:** 6

**Summary:**

The authors propose a method for exploiting repeated structure in the transition dynamics of an MDP.  Unlike prior works, they study symmetries present in the successor representation using graph automorphisms.  The full state space of the SR can be compressed by representing states within an automorphism group as a subgraph.  When computing the policy from the compressed SR, the rewards in a subgraph are averaged, which can negatively affect the policy. They show high compression on MDPs represented by Barabasi-Albert and Watts-Strogatz graphs, with minimal deviation from the uncompressed SR policy.

**Questions:**

- Is there a simple visual aid that could be added to convey the concepts of graph automorphisms or motifs in Section 3?
- In Section 4.1, it says that the MSE of the derived value function is dominated by a small number of nodes that are overly estimated.   Do the authors know why this is the case?
- Could the compression be performed on a partially learned successor representation?  Would one expect the discovered auto-morphisms to be persistent as the SR is learned?
- Are there any alternative compression methods that could be used to baseline the proposed approach?


**Limitations:**

The authors discuss the issue caused by pooling the rewards for states within a subgraph.  The authors could be more explicit about why they made certain decisions in Section 3 (for instance, only considering deterministic MDPs, and restricting motifs to two orbits in size).  Is this due to this approach's limitations, e.g., is finding motifs with more orbits computationally intractable?

**Recommended Decision:**

3: Accept

**Relevance:**

3: Solid fit

**Strengths And Weaknesses:**

Strengths:
- The idea of discovering repeated structure in the successor representation is novel.
- The proposed approach of compressing automorphisms in the SR achieves high compression on the MDPs on which they evaluated.
- The evaluation uses a large number of transition and reward functions, which makes the results more convincing.

Weaknesses:
- The authors do not state why compression of SRs is useful.  Compression could improve sample- and compute-efficiency while learning the SR, but it's not clear if this work could extend to partially learned SRs.  The last sentence of the conclusion does indicate that this is a possible future direction.
- In Table 1, the current evaluation metrics do not directly show the effect of the compression on the policy's performance.  An additional metric could be the difference in returns collected with the original policy and the compressed policy.


**Submission Track:**

Extended Abstract (4 Page)

---

### Official Review · Reviewer_P5Tu · 2022-10-14

**Confidence:** 4
**Soundness:** 3
**Presentation:** 2
**Contribution:** 2
**Overall Rating:** 5

**Summary:**

To the best of my understanding, the authors propose to find repeating structures in deterministic MDPs using standard graph analysis tools and then to compute successor representation over the abstracted MDP. The method is shown to be able to compress randomly generated graphs.

**Questions:**

* Do you assume to know the full MDP when constructing the graph motif (i.e. all states, actions and the full transition and reward functions)? How would the method differ if you only knew the environment from collected experiences?
* Could you re-state how successor representations fit into the graph-subgraph framework?

**Limitations:**

The limitations are stated clearly.

**Recommended Decision:**

2: Borderline

**Relevance:**

4: Highly relevant

**Strengths And Weaknesses:**

Since the results are from randomly generated graphs and no baseline results are reported, the contribution of the paper should come from the novel idea it proposes. However, what the paper proposes is not clear to me. I would be in favor of acceptance if the authors improve the explanation of their proposed method.

Comments:
* The connection between graph motifs and successor representation is not explained clearly.
* "If one were to sample trajectories from this graph following a policy, or compute a policy through sample-based policy improvement, the graph-subgraph structure would need to be represented hierarchically using the options framework" -- I am not convinced this is true. We *can* use the graph-subgraph structure together with options, but I don't think we *need* to. Either way, more explanation is needed.
* The second paragraph in Section 4 does not logically follow the first paragraph.
* The paper is missing a figure with an intuitive example of the kinds of local symmetries we are looking for. Even reproducing a figure from (6) would make the reading much easier.
* The paper could be more self-contained by describing graph motifs in greater detail.
* Minor comment: the paper uses different reference style to the other papers I reviewed.

**Submission Track:**

Extended Abstract (4 Page)

---

### Official Review · Reviewer_X3NJ · 2022-10-18

**Confidence:** 3
**Soundness:** 3
**Presentation:** 3
**Contribution:** 2
**Overall Rating:** 5

**Summary:**

The paper builds on top of the successor representation framework to compress the state transition matrix using a geometric decomposition of the graph. This leads to a more efficient representation of the problem which can allow for both compression and more efficient learning.

**Questions:**

Would benefit from a diagram clarifying the idea

**Limitations:**

Limited experiments.

A figure to illustrate the ideas of section 3 and 4 would be highly helpful.

**Recommended Decision:**

2: Borderline

**Relevance:**

4: Highly relevant

**Strengths And Weaknesses:**

Strentghts:
 - Clearly written
 - Simple idea combining the ideas of successor representation and the symmetry of complex networks

Weaknesses:
 - Would benefit from a diagram clarifying the idea
 - Experiment on real world complex networks would be good
 - Compression seems to be small in the experiment. The WS graph has almost the same size

**Submission Track:**

Extended Abstract (4 Page)

---

### Decision · Program_Chairs · 2022-10-21

Accept (Poster)